# Drug Resistance Mechanism of M46I-Mutation-Induced Saquinavir Resistance in HIV-1 Protease Using Molecular Dynamics Simulation and Binding Energy Calculation

**DOI:** 10.3390/v14040697

**Published:** 2022-03-28

**Authors:** Nilottam Rana, Atul Kumar Singh, Mohd Shuaib, Sanjay Gupta, Mahmoud M. Habiballah, Mustfa F. Alkhanani, Shafiul Haque, Mohd Salim Reshi, Shashank Kumar

**Affiliations:** 1Molecular Signaling & Drug Discovery Laboratory, Department of Biochemistry, Central University of Punjab, Bathinda 151401, Punjab, India; nilottam13@gmail.com (N.R.); singhatulkumar174@gmail.com (A.K.S.); mohd.shuaib9519@gmail.com (M.S.); 2Department of Urology, Pharmacology and Pathology, Case Western Reserve University, Cleveland, OH 44106, USA; gxs44@case.edu; 3Medical Laboratory Technology Department, Jazan University, Jazan 45142, Saudi Arabia; mhabibullah@jazanu.edu.sa; 4SMIRES for Consultation in Specialized Medical Laboratories, Jazan University, Jazan 45142, Saudi Arabia; 5Emergency Service Department, College of Applied Sciences, AlMaarefa University, Riyadh 11597, Saudi Arabia; mkhanani@mcst.edu.sa; 6Research and Scientific Studies Unit, College of Nursing and Allied Health Sciences, Jazan University, Jazan 45142, Saudi Arabia; shafiul.haque@hotmail.com; 7Toxicology and Pharmacology Lab., Department of Zoology, School of Biosciences and Biotechnology, Baba Ghulam Shah Badshah University, Rajouri 185234, Jammu & Kashmir, India; reshisalim@gmail.com

**Keywords:** drug resistance, saquinavir, M46I mutation, HIV-1 protease, anti-retroviral therapy

## Abstract

Drug-resistance-associated mutation in essential proteins of the viral life cycle is a major concern in anti-retroviral therapy. M46I, a non-active site mutation in HIV-1 protease has been clinically associated with saquinavir resistance in HIV patients. A 100 ns molecular dynamics (MD) simulation and MM-PBSA calculations were performed to study the molecular mechanism of M46I-mutation-based saquinavir resistance. In order to acquire deeper insight into the drug-resistance mechanism, the flap curling, closed/semi-open/open conformations, and active site compactness were studied. The M46I mutation significantly affects the energetics and conformational stability of HIV-1 protease in terms of RMSD, RMSF, Rg, SASA, and hydrogen formation potential. This mutation significantly decreased van der Waals interaction and binding free energy (∆G) in the M46I–saquinavir complex and induced inward flap curling and a wider opening of the flaps for most of the MD simulation period. The predominant open conformation was reduced, but inward flap curling/active site compactness was increased in the presence of saquinavir in M46I HIV-1 protease. In conclusion, the M46I mutation induced structural dynamics changes that weaken the protease grip on saquinavir without distorting the active site of the protein. The produced information may be utilized for the discovery of inhibitor(s) against drug-resistant HIV-1 protease.

## 1. Introduction

Human immunodeficiency virus type 1 (HIV-1) is a global epidemic with ~36.7 million already infected with the disease and ~1.8 million people being infected every year. More than half of the infected patients receive antiretroviral therapy (ART) [1]. Like other retroviruses, HIV-1 uses genetic economy to rely on a small genome size (10 kilobase) which ultimately produces proteins for viral particle formation, replication, and virulence. HIV-1 expresses polyproteins that are later on proteolytically processed into functional subunits. The HIV-1 protease is also responsible for the proteolytic processing of the polyprotein as an encoded process. The cleaved peptides subsequently undergo folding to from functional proteins, which in turn perform their role in viral life cycle. HIV-1 protease (HIV-PR) possesses two 99 amino acid monomers which form the catalytically active dimer form. Each monomer chain provides three residues, Asp25/25’, Thr26/26’, and Gly27/27’, which form the catalytic triad at the active site of the enzyme. Aspartate 25/25’ is the main catalytic amino acid residue and, thus, the protease is known as aspartate protease. The hydrogen bond network (known as the fireman’s grip) in the vicinity of protease active site provides relative rigidity to the enzyme core [2]. The amino acid residues 1–4 and 96–99 at the N- and C-termini are involved in the protease dimerization process, which is an important feature to achieve catalytic competence and the fine control of proteolytic cleavage of the polypeptide. An eleven (45–55) amino acid residue stretch (glycine rich) in each subunit of the protease folds into two anti-parallel β strands, known as the “flap” (Figure 1A). The flap forms a dynamic “roof” over the active site and shows conformational changes in response to substrate/inhibitor binding. Besides the catalytic aspartate, flaps, and four N- and C- terminal residues, the Arg8, Asp29, and Arg87 residues are involved in the interdomain interactions forms the dimer interface [3]. 

Saquinavir (SQ), the first FDA-approved protease inhibitor, is widely used in acquired immunodeficiency syndrome (AIDS) therapy. The first-generation anti-viral drugs, such as saquinavir, are peptidomimic of their respective processing substrate and, thus, competitively bind to the active site of the protease [4]. SQ tightly binds to the protease by non-polar van der Waals interactions through its side chains and interacts with both the active site and flap regions [5,6]. These interactions might be disrupted by mutations in the vicinity of the binding pocket of the protease, which ultimately produce drug resistance and decrease therapeutic outcomes. Drug resistance is a major problem in HIV-1 protease-inhibitor-based chemotherapy, which is characteristically associated with the decreased binding affinity of the inhibitor with the protease. HIV protease possesses a higher degree of structural flexibility and resistance to a variety of inhibitors by incorporating mutations in active and non-active sites. The non-active site mutations comparatively produce higher drug resistance. It has been shown that the protease can accommodate more than one mutation to increase the drug resistance for a given inhibitor without affecting the catalytic efficiency of the enzyme [7]. M46I is a non-active site mutation that occurs in the flap region of the protease. Recently, a clinical study on antiretroviral treatment-naïve patients in the Turkish population found the M46I mutation in high abundance [8]. Another study reported M46I as a major HIV-protease mutation in the Icelandic population [9]. Canducci et al., (2011) showed that a single mutation (M46I) imposed a 4.4-fold change in saquinavir resistance in mutated protease-infected CD4+ T cells in comparison to wild-type protease-infected cells [10]. The fold change in resistance was increased (up to ~7 and ~29) with the occurrence of one and two additional mutations, respectively. It has been reported that the M46I mutation decreases the SQ interaction (1.13 times) with the protease. The study also indicated M46I as one of the most prevalent mutations related to SQ treatment in HIV patients [4]. In a recent study, Yuan et al., (2021) studied the prevalence of HIV-1 drug resistance transmittance among treatment-naive individuals in China during 2000–2016. The study reported that individuals infected with a virus containing the M46I mutation in the protease accounted for increased rates of disease transmission [11]. It has been reported that the background mutations might affect the drug resistance potential of the primary mutation in HIV-1 protease. The study reported by Bastys et al., (2020) studied the effect of M46I and other background mutation(s) on the drug resistance potential of N88S and L76V and primary mutations in HIV-1 protease [1]. Based on recently published clinical facts (a high abundance or the most prevalent mutation associated with saquinavir treatment and a greater transmission potential) on M46I-mutation-mediated saquinavir resistance in HIV patients, we considered only the M46I mutation (primary mutation) in the present study [4,8,9,11]. Taken together, there is an urgent need to study the effect of mutation on the binding of inhibitor(s) to the HIV-1 protease. The detailed mechanism of M46I mutation in the SQ–protease interaction has not yet been established. The present study was designed to provide an insight into the energetics and structural conformation changes in WT (wild-type protease), SQ-WT (saquinavir-bound wild-type protease), MI (M46I mutation-carrying protease), and SQ-MI (saquinavir-bound M46I protease) systems in a compressive mode by performing a molecular dynamics simulation approach.

## 2. Results and Discussion

### 2.1. Molecular Docking and Secondary Structure Analysis 

The wild-type and M46I-mutation-carrying HIV-1 protease Pdb structures were overlapped, and the results are shown in Figure 1B. The results indicate that most of the protein structure, including the active site of the protein were not changed. It should be noted that the minor change in the flap region was obeserved in the mutated structure. The secondary structure of the WT, SQ-WT, SQ-MI, and MI system were studied using the trajectory obtained after 100 ns of MD simulation. Quality control parameters and secondary structure analysis for the test system are shown in Appendix A. The secondary structure analysis showed that the M46I mutation significantly impacted the β-sheet structures in HIV-1 protease. Saquinavir binding further induced the changes in 5′ Helix of the MI protease which were not found in SQ-WT system. The type of interaction and amino acids involved in wild-type/mutated HIV-1 protease–saquinavir complex is depicted in Figure 1C,D respectively. Further, the molecular docking experiment revealed −12.2 and −11.16 docking scores for the saquinavir-bound wild-type and M46I-mutation-carrying HIV-1 protease, respectively. The wild-type complex showed an increased number of van der Waal interactions in comparison to the saquinavir-mutated protease complex. The results indicate that the M46I-mutation-mediated structural change in the secondary structure of the flap region of the protein decreased the HIV-1 protease binding potential of saquinavir by lowering the van der Waals interaction. 

### 2.2. Effect of the M46I Mutation on RMSD of the Backbone Atoms 

Conventional molecular dynamics (MD) simulations were conducted on four systems, namely, wild-type HIV-1 protease (WT), saquinavir-bound wild-type HIV-1 protease (SQ-WT), M46I-mutation-carrying HIV-1 protease (MI), and saquinavir-bound M46I-mutation-carrying HIV-1 protease (SQ-MI). The MD simulation was performed for 100 ns on each system. 

Figure 2A shows the root mean standard deviation (RMSD) of the backbone atoms (N, aplha C, and carbonyl C). The results showed that the WT and SQ-WT systems possess highly stable RMSD of the backbone atoms throughout the simulation period. The MI and SQ-MI systems did not show significant differences in RMSD with the wild-type bound and un-bound systems during the initial simulation period (1–5 ns). After that, the difference in RMSD started increasing and was in the range of 0.3–0.75 Å throughout the simulation period. It is noteworthy that the MI system RMSD was relatively higher than the WT/SQ-WT system but significantly lower than the SQ-MI system, especially during 10–50 ns time period of simulation. Further, during 60–70 ns and 85–95 ns, MI and SQ-MI showed an overlapping RMSD pattern. The results indicate that the M46I mutation increases the RMSD of the backbone atoms in WT HIV-1 protease, which was further increased due to binding of an inhibitor molecule to the mutated protein. It is important to note that due to theM46I mutation the WT HIV-1 protease assembly was deviated from its original structure. Furthermore, the binding of saquinavir (active-site-targeted inhibitor) produced additional deviation in the protein structure during most of the MD simulation period. Other studies also reported the increased RMSD due to mutation in the flap region of HIV-1 protease, although the reported differences were smaller. This might be due to the shorter time of the MD simulation [12]. 

### 2.3. Effect of the M46I Mutation on the Fluctuation of Amino Acid Residues 

Besides RMSD, the RMSF (root mean square fluctuations) is another important measure of the structural fluctuations in the protein structure during the MD simulation. Like RMSD, RMSF indicates the fluctuation in whole protein or significant displacements of a small structural portion of the test system. RMSF provide clear-cut fluctuations in particular/group of atoms during the simulation period. Figure 2B shows the RMSF of amino acid residues in WT, SQ-WT, MI, and SQ-MI systems during the 100 ns simulation period. The results showed two regions (45–55 and 145–155 amino acid residues) possessing maximum fluctuations. The 45–55 and 145–155 amino acid residue stretches constitute the “flap region” of the HIV-1 protease of two identical subunits. The MI and SQ-MI systems showed higher fluctuation (0.25–0.75 nm) in comparison to the WT/SQ-WT systems. It should be noted that saquinavir decreased the fluctuation in the flap region after binding with the WT/MI proteins almost to a similar extent. The results showed that the M46I mutation did not significantly affect the overall structure of the protein, except the flap region. Our results are in the line with the previous study which showed the fluctuation in the flap region of the HIV-1 protease during the 22 ns simulation period [12]. Previously, it has been mentioned that the mutation in the HIV-1 protease flap region results in its increased opening in terms of both time and space in comparison to the wild-type protease. In our study, larger fluctuations in the flap region of the M46I-mutated protein indicate a comparatively increased opening conformation. Flap region dynamics are supposed to regulate the binding and dissociation of drugs to HIV-1 protease [13]. Thus, it might be inferred that the M46I mutation results in increased dynamics at the flap region, which might affect the protease-binding efficacy of saquinavir. Molecular docking also revealed a saquinavir interaction with the active site and the flap region of the protease.

### 2.4. Effect of the M46I Mutation on Conformational Compactness

The radius of gyration (Rg) and solvent accessible surface area (SASA) parameters were studied in the WT, SQ-WT, MI, and SQ-MI systems. The trajectories obtained from the MD simulation were analyzed for Rg and SASA for the entire 100 ns of the simulation period. Figure 2C depicts the variation in Rg among the studied systems. The WT and SQ-WT systems did not show much difference in the Rg pattern and were in range of 1.71–1.75 nm. Conversely, the MI and SQ-MI systems showed Rg fluctuations that were in the range of 1.68–2.04 nm during the MD simulation period. It should be noted that due to the mutation the protease showed low Rg values in comparison to the unbound wild-type structure during the initial MD simulation period (1–5 ns). After that (up to 10 ns), it increased and stabilized the complex up to 50 ns and further increased significantly. In the last 35 ns of the simulation period (75–100 ns), the Rg value of the M46I structure reached close to the SQ-WT structure. Interestingly, the binding of saquinavir to the M46I protease increased the Rg value during the initial MD period (1–5 ns) in comparison to the unbound M46I structure. The Rg value was markedly increased from the 12–50 ns period and was significantly higher than the unbound M46I structure. Following the similar pattern (among WT and MI), the Rg value of SQ-MI tended to be lower and reached the MI level at several places during the 55–100 ns simulation period. Rg indicates the fluctuation/conformational compactness of a test protein, which is based on the moment of inertia calculated for the carbon alpha atoms from its center of mass. Increased Rg values define increased conformational flexibility in the test structure. Here, in the MD simulation study, unbound wild-type HIV-1 protease showed a more stable conformational flexibility (lesser Rg value) than saquinavir-bound protease. The slightly increased conformational flexibility in the saquinavir-bound system might be due to the motion in the flap region, which is a known phenomenon in drug/inhibitor bound proteases. Moreover, fluctuation in the Rg value of the mutated protease during the simulation period also indicates that, at some places, the structure became more rigid, while at other places it showed less compactness. Overall, the change in the Rg value of the mutated structure in comparison to the wild-type structure was in the line of mutation-based conformational distortion hypothesis, which ultimately results in a lose grip on the drug molecule [14].

The SASA values for all the studied system varied between 102–221 nm during the MD simulation (Figure 2D). The results showed that unbound wild-type HIV-1 protease possessed a lower SASA value than the saquinavir-bound structure. The SASA value of the WT system was significantly lower during the 28–32, 42–56, and 90–100 ns simulation periods. Up to the 20 ns period, no significant changes between the SASA value of the WT and SQ-WT systems were recorded, but, after that, the difference was quite significant throughout the simulation period. It should be noted that the SASA of the WT structure was significantly decreased up to 105–102 nm during the last few nanoseconds of MD simulation. After that, the SASA value was increased in the saquinavir-bound wild-type structure and concentrated around 110 nm during the rest of the simulation period. Conversely, the SASA value among MI and SQ-MI showed much difference among themselves. During the entire simulation period, the saquinavir-bound/unbound mutated structure showed a higher SASA (concentrated around 115 nm) value throughout the simulation period. The SASA value of the unbound mutated structure was concentrated around 110 nm. The surface of test protein, which is accessible to the surrounding solvent molecules, is known as the solvent-accessible surface area (SASA). The solvation event is an important feature which plays an essential role in the folding of the protein and its stability. It also affects the interaction among proteins as well as its structural modification. Besides, the SASA value also provides valuable information regarding the hydrophobic compactness of the protein. The solvation effect can be calculated in terms of SASA in the MD simulation study [15]. An increased SASA value in the M46I HIV-1 protease structure in comparison to the wild-type protease might be due to the conformational flexibility in the protein. Changes in the SASA value indicated that M46I decreased the hydrophobic compactness of the protein, which was further increased after the binding with saquinavir.

### 2.5. Effect of the M46I Mutation on the Hydrogen Bond Formation Pattern

Hydrogen bond formation is one of the important non-covalent interactions during protein-ligand formation. With other non-covalent interactions (such as van der Waals contacts and electrostatic forces) it allows an understanding of the binding potential of the ligand to its target protein. The hydrogen bond (H-bond) formation in the test protein (WT, SQ-WT, MI, and SQ-MI) systems was calculated during the entire simulation period. Figure 3A,B depicts the hydrogen bond formation within the protein (intra-molecular) and between the protein and the surrounding water molecules (inter-molecular). The results showed that the average H-bond formations in unbound and saquinavir-bound wild-type proteases were 126.60 and 129.5, respectively. Comparatively, saquinavir-bound mutated protease depicted a lesser number of intra-molecular H-bond formation (~124.32) than the WT, SQ-WT, and MI systems (Figure 3A). The entire H-bond pattern among the test system was opposite in the case of inter-molecular H-bond formation (Figure 3B). The average H-bond formations at the binding site of the protease in the SQ-WT and SQ-MI complexes were 0.672 and 0.535, respectively, during the 100 ns MD simulation period (Figure 3C,D). The H-bond pattern generated in the present study indicate that the mutation decreased the rigidity in the protease structure. Increased intra-molecular H-Bond formation in the saquinavir-bound wild-type protease in comparison to the corresponding unbound structure indicates that the binding of the ligand provides rigidity to the protein structure. Conversely, the decreased H-bond pattern in the SQ-MI complex indicates that the M46I mutation inversely alters the structural rigidity of the protease upon saquinavir binding. Decreased H-bond formation at the ligand binding site in SQ-MI protease in comparison to the SQ-WT complex further indicates the weak interaction between the inhibitor and binding site. Overall the H-Bond pattern indicates that the M46I mutation decreased the rigidity of the protein structure and the strength of saquinavir binding to HIV-1 protease.

### 2.6. Effect of the M46I Mutation on Flap Curling and Opening

The distance between Asp25-Ile50 and Asp-25-Ile149 residues was calculated to obtain insight into the role of M46I in the flap curling of the protease. The angle between the three adjacent carbons of the G48–G49–I50 amino acid residues (also known as TriCa) was studied to assess the prevelance of closed/semi-open/open conformations in the test protein systems [12,16]. These parameters were studied in the WT, SQ-WT, MI, and SQ-MI systems throughout the MD simulation period. To map the distance between the test amino acid residues, the MD trajectory was used to extract the Pdbs at 1, 20, 40, 60, 80, and 100 ns. The distance was calculated in the extracted Pdbs using PyMol software (Appendix A). Figure 4A,B depicts the angle between the three adjacent carbons of the G48–G49–I50 amino acid residues (also known as TriCa) in the WT/MI and SQ-WT/SQ-MI systems, respectively, during the entire 100 ns simulation period in the test systems. The TriCa angles in the SQ-WT/SQ-MI systems depicted more fluctuation than the respective wild-type systems. Next, we analyzed the probability of the TriCa angle across the simulation period in all the test systems, and the results are depicted in Figure 4C,D. Figure 4C compares the angle probability in unbound WT and mutated structures. The results showed that the M46I mutation caused the higher probability (0.07) for 144–148 degree angels in comparison to the WT structure where 146 degree angles showed a higher probability (Figure 4C). The saquinavir-bound M46I-mutation-carrying protease shifted the angle axis towards the right side of the plot (Figure 4D). Figure 4E depicts the variation in Asp25-Ile50 distance during the 100 ns simulation in all the test systems. The unbound wild-type protease showed little fluctuation in the Asp25-Ile50 distance throughout the 100 ns simulation period. The SQ-WT system showed a comparatively stable distance fluctuation compared to the WT system, especially after the 40 ns period. The MI system showed an increased distance fluctuation in comparison to the WT and SQ-WT systems but a smaller fluctuation than the SQ-MI complex. The average distance between thee Asp25 and Ile50 residues in the WT, SQ-WT, MI, and SQ-MI systems were 1.4359 ± 0.1914, 1.5600 ± 0.1306, 1.7026 ± 0.2291, and 1.8856 ± 0.2939, respectively. For visualization purposes the snapshot of the structures at 1, 60, and 100 ns are shown Appendix A. Figure 4F depicts the variation in the Asp-25-Ile149 distance during the 100 ns simulation in all the test systems. During the initial few nanosecond simulation periods (0–5 ns) the mutated unbound (MI) system showed a minimum Asp-25-Ile149 distance among the test systems. After that, the distance in the MI system was maintained throughout the simulation at a higher level. During the 20–100 ns simulation period, the fluctuation in the Asp-25-Ile149 distance was in the sequence of WT < SQ-WT < MI. The SQ-MI system showed a higher Asp-25-Ile149 distance fluctuation among all the test systems and was slightly lower than the MI system. Interestingly, the Asp-25-Ile149 distance fluctuation in the SQ-MI system covered all the fluctuation ranges shown by the other three test systems during the simulation period. The average distance between the Asp-25 and Ile149 residues in the WT, SQ-WT, MI, and SQ-MI systems were 2.0348 ± 0.3014, 2.0923 ± 0.2797, 1.7312 ± 0.1398, and 1.3359 ± 0.1837, respectively. Further, for visualization purposes, the snapshots of the structures at 1, 60, and 100 ns are depicted in Figure 4.

The curling of HIV-1 protease flaps is an important phenomenon observed during its conformational changes (closed, semi-open, and fully open). It has been reported that the curling of the flap is coupled to the motion of the entire flap backbone structure, which results into the semi-open to fully open conformation of the protease [17]. The distance between the Asp25 β-Carbon and Ile50 α-Carbon in the same chain has been widely used as an important index to show the flap states [18]. The stable Asp25-Ile50 distance in the SQ-WT system indicates that the binding of saquinavir stabilized the flap movement conformation in comparison to the WT system. In this situation, the movement of the flap and its opening has been blocked, which indicates the protease inhibition potential of the drug. The increased average Asp25-Ile50 distance in MI and SQ-MI (in comparison to WT and SQ-WT) indicates that the M46I mutation increased the curling of the flap. A decrease in the Asp25-Ile50 distance up to the non-mutated system levels at most of the simulation period indicates an inward (towards core of the active site) curling of the flap in the M46I mutation systems. In comparison to WT protease, saquinavir was unable to stabilize the flap motion in M46I protease. It indicates that the mutation nullifies the saquinavir-binding-mediated stabilization of HIV-1 protease flap dynamics. The Asp25 β-Carbon and Ile50 α-Carbon in the 1HHP.pdb crystal structure of apo HIV-1 protease (1.72 nm) has been considered as the semiopen conformation. Thus, any snapshot with a distance ‘<’, ‘=’, or ‘>1.72 nm’, was considered in closed, semi-open, and open conformation, respectively [19]. Similarly, the distance between the Asp25 β-Carbon and Ile50 α-Carbon for the open and closed conformation structures of HIV-1 protease were caluculated by utilizing the 1TW7 and 4EJK PDBs. The open and closed HIV-1 protease structures depicted 1.88 and 1.37 nm distances between the residues, respectively. Based on these facts, our results indicate that WT and SQ-WT showed closed-flap conformation (<1.72 nm), and the M46I mutation stabilized the semi-open conformation (~1.72 nm), whereas the saquinavir-bound mutated structure depicted the open conformation (>1.72 nm Asp25-Ile50 distance) during the simulation period.

### 2.7. Effect of the M46I Mutation on Active Site Compactness

The distance between the Asp25-Asp124 radiuses were calculated to study the active site compactness in the WT, SQ-WT, MI, and SQ-MI systems throughout the MD simulation period [16]. To map the Asp25-Asp124 distance, the MD trajectory was used to extract the Pdbs at 1, 20, 40, 60, 80, and 100 ns. The distance was calculated in the extracted Pdbs using PyMol software (Appendix A). Figure 5A depicts the fluctuation in the Asp25-Asp124 residue distance during the entire simulation period in the test systems. The unbound and saquinavir-bound wild-type protease (WT and SQ-WT) did not show any significant fluctuation in the Asp25-Asp124 distance throughout the 100 ns simulation period. The MI system showed a slightly increased distance during the 55–75 and 90–95 ns simulation periods in comparison to the WT and SQ-WT complexes. Although the SQ-MI system possessed a comparatively higher fluctuation for the Asp25-Asp124 distance among the test systems, the fluctuation led to a minimized level of non-mutated systems throughout the simulation period (Figure 5A, Panel I and II). The average distance between the Asp25 and Asp124 residues in the WT, SQ-WT, MI, and SQ-MI systems were 0.7843 ± 0.0715, 0.6838 ± 0.0338, 0.6659 ± 0.0215, and 0.6529 ± 0.02235, respectively. Further, for visualization purposes the snapshots of the structures at 1, 60, and 100 ns are depicted in Figure 5, panel II.

To study the compactness/size of the active site, we calculated the distance between Asp25 and Asp124 as described earlier [18]. The Asp25 residue belongs to chain A, and Asp124 is the same aspartate residue in chain B. They are involved in the HIV-1 protease active site triad formation. The Asp25-Asp124 distance was taken as a parameter to predict the compactness of the active site in the presence/absence of saquinavir in M46I-mutation-carrying proteases [18]. The absence of a significant alteration in the Asp25-Asp124 distance in saquinavir-bound/unbound wild-type protease indicates that the active site was stable and compact. The increased distance in the saquinavir-unbound/bound M46I-mutation-carrying proteases indicates that the active site was less compact in comparison to the non-mutated systems.

### 2.8. Effect of the M46I Mutation on Conformational Dynamics and Stability

In order to gain insight into the dynamic states attained by HIV protease in both WT and MI, the principal component analysis (PCA) was performed. The differences in the dynamic states attained by the test systems were studied by projecting their trajectories onto a two-dimensional space [20,21]. The red and blue dots in the PCA plot are representative of two states, while the white dots represent intermediate states (Figure 6, Panel I). The transition from the red to blue color indicates the periodic jump between the two states. The PCA revealed that first three principal components (PCs) acquired the most of data variance and can be utilized to explore the dynamic behaviour of the test systems. In WT and MI protease, the first three PCs acquired a total of 54.6% (28.13%, 17.31%, and 9.13% of data variance for PC1, PC2, and PC3, respectively) and 76.4% variance (61.72%, 11.11%, and 3.53% of data variance for PC1, PC2, and PC3, respectively), respectively. Similarly, the PC-acquired data variance in the SQ-WT and SQ-MI systems were 43% and 64%, respectively (Figure 6 Panel I). The PCA results indicate that the M46I mutation significantly increased PC1 contribution, which might correlate with the increased flap movement in the protease. The binding of saquinavir did not alter the PC1 contribution significantly in WT proteases, but in the MI system, a lesser PC1 contribution was observed (Figure 6, Panel I). This indicates that the M46I mutation decreased the effect of saquinavir binding on the alteration of the PC1 contribution. Moreover, the projection of PC1 onto the RMSF also indicated the significant contribution of PC1 in the total fluctuation of the HIV protease structure (Appendix A). The DCCM analysis results showed that the M46I mutation significantly increased both the correlated and anti-correlated motions in HIV protease in comparison to the WT structure. Saquinavir binding reduced the motions in the WT protease but showed no significant effect on the MT protease (Appendix A). Thus, it might be inferred that the binding of the inhibitor in the mutated protein was not able to modulate the flap motion negatively. Free energy landscapes (FEL) were generated to understand the energetically stable conformation(s) among the test systems. The SQ-WT system showed an increased local minimum energy basin in comparison to the unbound WT system (Figure 6, Panel II). In the M46I-mutation-carrying protease, the binding of saquinavir was unable to maintain the minimum energy basin in comparison to the SQ-WT system. The results indicate that the binding of saquinavir produced a lesser local minimum energy basin, which might be due to the increased motion in the MI protease. The FEL results also indicate that the M46I mutation provided increased conformational flexibility to the saquinavir-bound mutated protease. The FEL results are in the line with the RMSD, RMSF, and flap curling/open conformation, where the SQ-MI system showed increased fluctuation in comparison to the SQ-WT system.

### 2.9. Effect of the M46I Mutation on Saquinavir-HIV-1 Binding Energetics

The comparative interaction potential of saquinavir with the wild-type (SQ-WT) and M46I mutation-carrying (SQ-MI) HIV-1 protease was studied in terms of the binding free energy calculation. The binding free energy was calculated for the last 10 ns time period of the MD simulation using MM-PBSA calculations (Figure 7, Table 1). The average binding energy of the SQ-WT and SQ-MI complexes was −186.390 and −150.657 kJ/mol, respectively (Figure 7A and Table 1). The comparative energy contributions of all the 198 amino acid residues of the WT and MI HIV-1 protease complexed saquinavir are shown in Figure 7B,C respectively. Further, the binding free energy was decomposed into its major contributors (van der Waals, electrostatic, polar solvation, and SASA energy), and the results are shown in Table 1. Moreover, to explore the role of M46I mutation in saquinavir resistance in detail, the binding energy was decomposed per amino acid residue involved in the SQ-WT and SQ-MI complex formation, and the residues whose contribution were more than ±1 are depicted in Figure 7D,E, respectively. The result showed that 10 amino acid residues (Asp29, Lys45, Arg87, Lys119, Asp124, Lys142/144/154, and Arg156/186) contributed to the positive energy in the SQ-WT complex. Asp124 showed the maximum positive energy contribution in the wild-type complex (Figure 7D). A total of 31 residues contributed toward negative energies, and Ile47 contributed the most (Figure 7D). Conversely, the SQ-MI system showed the involvement of only 6 and 17 amino acid residues toward the positive and negative energy contributions, respectively (Figure 7E). Moreover, Figure 7B,C indicate that due to mutation saquinavir is not able to interact with the amino acid residues present in chain B (amino acid residues 100–198). Furthermore, Figure 7D,E also indicate the lesser involvement of chain B amino acid residues (contributing > −1 kJ/mole) in drug binding.

The binding free energy of the complex in the SQ-MI complex indicated about a 20% decrease (Table 1) in the energy, which might be responsible for the lesser anti-viral therapy outcome in the patients on saquinavir treatment carrying the M46I HIV-1 protease mutation. Further, the binding free energy was decomposed into its major contributors (van der Waals, electrostatic, polar solvation, and SASA energy), and the results are shown in Table 1. Previously it has been reported that the van der Waals interaction plays a major role in inhibitor/natural substrate binding to HIV-1 protease [22]. In our study, we found that the van der Waals interaction played a putative role in binding free energy of saquinavir-HIV protease, followed by SASA and electrostatic contributions. A significant decrease in van der Waals contribution in the SQ-MI complex indicates an important role of the M46I mutation in decreasing the interaction between saquinavir and HIV 1 protease binding. The involvement of the van der Waals interaction in the binding of saquinavir was in agreement with the previous report on other mutation-carrying HIV-protease [23]. It should be noted that the extent of the electrostatic and SASA energy contributions was found to be different with the published report. These data indicate that the M46I mutation differentially affects the energetics of saquinavir-HIV 1 protease in comparison to other known mutations. The large number of amino acids involved in the interaction/energy contribution with the two opposite sides of the cavity i.e., the flap (Ile47, Gly48, Gly49, Ile50, GLy51, Phe53, Ile54, Ile146, Gly148, Ile149, and Ile153) and the active site (Asp25, Gly27, and Gly126) region indicates the potential inhibitory action of SQ. However, the M46I mutation altered this trend and highly reduced the significant involvement of energy contributory amino acids in the flap and active site regions of the protease (Figure 7D,E). The amino acid contribution energy plot (Figure 7D,E) also indicated that the saquinavir interaction with the active site amino acid residues (Asp25, Thr26, Gly27, Asp124, Thr125, and Gly126) was decreased due to the M46I mutation.

## 3. Materials and Methods

### 3.1. Retrieval and Prepration of the Receptor and Ligand

The three-dimensional structure of HIV protease was obtained from the RCSB protein data bank (PDB-ID:5YOK) [24,25]. The downloaded structure of HIV protease had a resolution of 0.85 Å and was considered to be a wild-type (WT) protease. The three-dimensional structure of saquinavir was obtained from the PubChem database [26]. The downloaded three-dimensional structure of HIV protease was loaded into the PyMol software, and all water molecules were removed. To introduce the M46I mutation, the mutagenesis module of PyMol software was utilized [27,28]. The protease containg the M461 mutation was considered to be a mutated (MI) protease. The downloaded three-dimensional structure of HIV protease had inhibitor (KNI-1657) bound with it. The PDB structures of both the WT and MI HIV protease containing the co-crystallized inhibitor KNI-1657 were loaded into the protein preparation wizard of the Schrodinger software package. In the preprocessing step, the protein preparation wizard of the Schrodinger software package checked the PDB structure for missing side chains, alternative positions, bond orders, and corrected them. Further, in the optimization step, hydrogen bonds present in the PDB structure were optimized, and, finally, the PDB structures of both HIV proteases were energy-minimized using the optimized potential of liquid simulations 3 (OPLS3) force field [29]. The three-dimensional structure of saquinavir obtained from the PubChem database was loaded into the LigPrep wizard of the Schrodinger software package for ligand preparation. The LigPrep module of the Schrodinger package converts two-dimensional structures of ligands into three-dimensional structures, adds hydrogens, and performs energy minimization on ligand molecules. In the present study we considered the continous numbering of amino acid residues in the dimer structure. The 99 amino acids of Chain A and B are numbered as 1–99 and 100–198, respectively.

### 3.2. Receptor Grid Generation and Molecular Docking

The prepared structures of the WT and MI HIV proteases, obtained from the protein preparation wizard, were utilized to generate a receptor grid on the basis of the bound inhibitor. The bound inhibitor present in the receptor was used to generate the receptor grid. The grid box (20 Å × 20 Å × 20 Å) was generated on coordinates X-15.91, Y-21.9, and Z-16.56 for WT along with X-15.83, Y-21.9, and Z-16.56 for MI. Ligand docking was performed using the Glide module of the Schrodinger software package [30,31]. The prepared ligand was docked onto the previously generated receptor grid. Glide uses an Emodel scoring function to compare protein ligand docked poses and Glide scores. For the analysis of docking results, docked files of receptors and ligands were exported from the Schrodinger package in .pdb format and combined together to form the docked protein–ligand complex. Interacting residues of both the WTand MI HIV protease with the ligand were identified using BIOVIA Discovery Studio, and representations were prepared using PyMol and BIOVIA Discovery Studio (https://discover.3ds.com/discovery-studio-visualizer-download, https://pymol.org/2/).

### 3.3. Molecular Dynamics Simulations

Molecular dynamics (MD) simulations are widely used to understand the stability and dynamic behaviour of proteins and protein–ligand complexes [32,33,34,35]. MD simulations were carried out for unbound WT and MI protease and complexed with saquinavir seperately. MD simulations were perfomred using WebGRO for Macromolecular Simulations (https://simlab.uams.edu/). The server uses the GROMACS MD simulation package to perform the MD simulation of protein and protein–ligand complexes [36]. The topology and parameter for the protein were generated using GROMOS 54A7 force field [37]. The topology and parameter for the ligand were generated using PRODRG server [38]. Each protein and protein–ligand complex was placed inside a cubic simulation box with 1 Å distance to the edges. Solvation was perfomred using a single point charge (SPC) water model and counter ions (NaCl) were added to achieve a molarity of 0.15 M. Each system was energy-minimized using a steepest descent method, followed by a short (500 ps) equilibration in an NVT ensemble and a subsequent 500 ps in an NPT ensemble [39]. The temperature and pressure of each system were set at 300 K and 1 bar, which was controlled by a Berendsen thermostat and a Parrinello–-Rahman barostat, respectively [40,41]. The integration step of 2 fs was used. Each system was simulated for 100 ns, and the snapshots were saved every 10 ps for further analysis. The resulting trajectories were analyzed using various analysis tools provided with the GROMACS package [36]. Several structural parameters were measured, such as root-mean-square deviation (RMSD), the radius of gyration (Rg), solvent-accessible-surface-area (SASA), root-mean-square-fluctuation (RMSF), hydrogen bonds, secondary structure, angles, and distances. The trajectories were visualized using Visual Molecular Dynamics (VMD) and the graphs were prepared using Grace software (https://plasma-gate.weizmann.ac.il/Grace/).

### 3.4. Principal Component Analysis and Dynamic Cross-Correlation Matrix (DCCM)

Principal component analysis (PCA) is a widely applied technique to identify the patterns in high-dimensional data. A PCA was performed on the resulting trajectories obtained from the MD simulation experiment. The principal components (PCs) in MD simulation trajectories highlight the concerted atomic displacements of biomolecules. These concerted atomic displacements describe the major conformational changes in the structure of biomolecules. Mathematically, PCs were determined by the diagonalization of data covariance matrix C:C = VΛV^T^
(1)
which provided the diagonal matrix Λ with the eigenvalues as diagonal entries and matrix V with the corresponding eigenvectors [42]. In order to perform the PCA, GROMACS MD simulation trajectories (.xtc) were converted into nanoscale molecular dynamics format (NAMD) format (.dcd) using VMD software. The Bio3D package developed by Grant et al., (2006), was used for the PCA. Bio3D is an R package that is widely used for MD simulation data analysis. Furthermore, we performed the dynamic cross-correlation matrix (DCCM) analysis to observe the time-correlated motions of the protein as described elsewhere [43]. The calculation of the DCCM is performed by the equation:C_ij_ = {Δr_i_*Δr_j_}/({Δr_i_^2^} {Δr_j_^2^})^½^
(2)
where Δri and Δrj represent the movement of atoms i and j from their mean position and C_ij_ represents the matrix. The DCCM was calculated using the Bio3D package.

### 3.5. Free Energy Landscape

Free energy landscapes (FEL) are widely used to identify the lowest energy conformations achieved by the biomolecules in MD simulation studies. The deep valleys in FEL represent the stable low-energy conformations, while the boundaries between deep valleys represent the intermediate conformations achieved by the biomolecules throughout the MD simulation. The two PCs obtained from the PCA were used for the generation of FEL with the help of following equation:ΔG (PC1, PC2) = −KBTlnP (PC1, PC2)(3)

In above equation, PC1 and PC2 are reaction coordinates, KB is the Boltzmann constant, and P(PC1 and PC2) is the probability distribution function of the system along the first two PCs. The changes in enthalpy (ΔH), standard free energy (ΔG), and entropy (ΔS) were calculated using the following equation:ΔG = ΔH − TΔS(4)

In above equation, ΔH represents enthalpy, T represents the temperature in Kelvin, ΔS represents the entropy of the system, and ΔG is the Gibbs free energy [44]. For the generation of FEL, PCs were calculated using the GROMACS analysis tools *gmx_covar* and *gmx_anaeig.* The *gmx_sham* tool was used to calculate the FEL.

### 3.6. Binding Free Energy Calculation

The molecular mechanics/Poisson–Boltzmann surface area (MM-PBSA) was calculated from the obtained MD trajectories in order to illustrate the structural stability of the protein–ligand complex and the molecular interactions of the ligand with the binding site of the WT and MI protease. The MM-PBSA analysis calculates a robust estimation of free-binding energy, contacts between the protein and ligand atoms, and the subsequent effect of the surrounding solvent on the binding affinity of ligands [45,46]. The binding free energy is expressed as:ΔG_binding_ = G_complex_ − (G_protein_ + G_ligand_)(5)

In above equation, G_complex_ is the total free energy of the protein–ligand complex, G_protein_ is the free energy of the protein, and G_ligand_ is the free energy of the ligand. Leaving the entropy terms, the free energy for each single species can be calculated as:ΔG = ΔE_MM_ + ΔG_solv_(6)
where ΔE_MM_ is the gas-phase average molecular mechanics interaction energy change influenced by the ligand binding. ΔG_solv_ is the solvation free energy change influenced by the ligand binding. E_MM_ encompasses the electrostatic, bonded, and van der Waals interaction energy, and, on the other hand, G_solv_ encompasses the polar and non-polar solvation energy terms.

## 4. Conclusions

In the recent study, we investigated the role of the M46I mutation on the binding of saquinavir (SQ) and HIV-1 protease. The inhibitor interacts with the flap region and the active site of the protein, mainly through van der Waals and hydrogen bond interactions. The M46I mutation produced an alteration in the β-sheet (near the flap region) structure of HIV-1 protease. Moreover, SQ binding additionally altered the 5′ helix of the mutated protease. The molecular dynamics study revealed that the mutation caused increased movement in the flap region and comparatively stabilized the open conformation of the protease after inhibitor (SQ) binding. The study revealed that the active site of the protease was minimally affected by the mutation. Further, the parameters such as RMSF, TriCa angle, Asp25-Ile50/Ile149 distance, DCCM, and the PCA analysis revealed increased fluctuation/motion in the SQ-bound mutated protease in comparison to the wild-type structure. Furthermore, the SQ-MI complex showed significantly less binding energy than the SQ-WT protease complex. The significant decrease in van der Waals energy contribution for the total binding energy composition in the mutated protease-SQ complex revealed loose binding of the inhibitor. Overall, the results indicate that the M46I mutation largely affects the dynamics of the flap region of the protease, which leads to the decreased interaction between mutated protease and saquinavir.

## Figures and Tables

**Figure 1 viruses-14-00697-f001:**
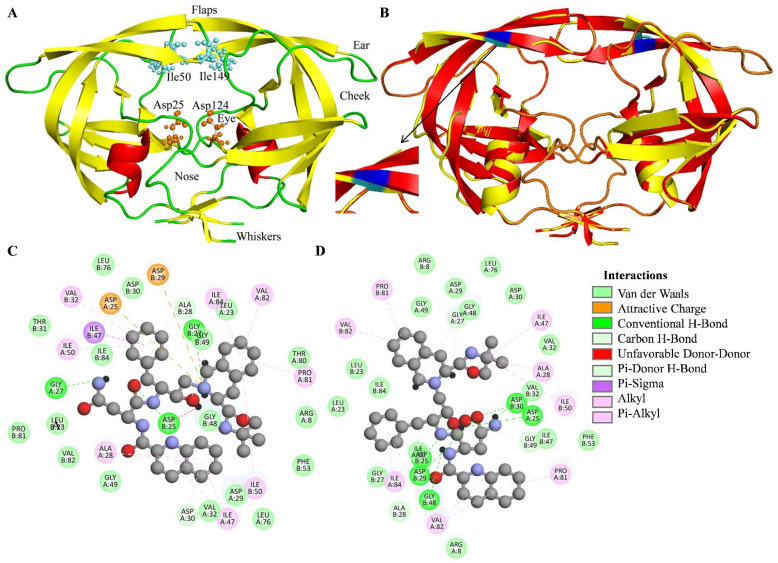
Depiction of HIV-1 protease native/mutated structures and saquinavir interaction. (**A**) Cartoon representation of native HIV-1 protease, showing the catalytic (orange color) and flap terminal residues (green color). The red, yellow, and green colors reperesent helix, β-sheet, and loop structures, respectively. (**B**) Overlapped cartoon representation of native (red) and M46I-mutation-carrying (yellow) HIV-1 protease. The blue and cyan colors represent methionine and isolecucine residues, respectively. (**C**) Residue interaction between saquinavir and wild-type HIV-1 protease and (**D**) residue interaction between saquinavir and M46I-mutation-carrying HIV-1 protease.

**Figure 2 viruses-14-00697-f002:**
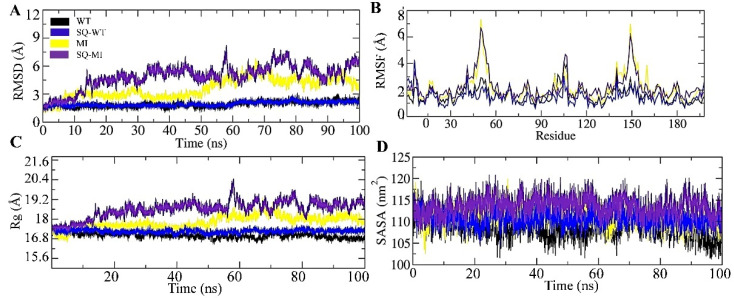
Effect of the M46I mutation on RMSD, RMSF, Rg, and SASA of saquinavir-bound HIV-1 protease. (**A**) RMSF (**B**) RMSF, (**C**) Rg, and (**D**) SASA of WT, SQ-WT, MI, and SQ-MI systems during the 100 ns simulation. The black, blue, yellow, and purple colors represent the WT, SQ-WT, MI, and SQ-MI systems. WT, Wild-type; SQ-WT, Saquinavir bound wild-type; MI, M46I-mutation-carrying protease; and SQ-MI, Saquinavir-bound mutation-carrying protease.

**Figure 3 viruses-14-00697-f003:**
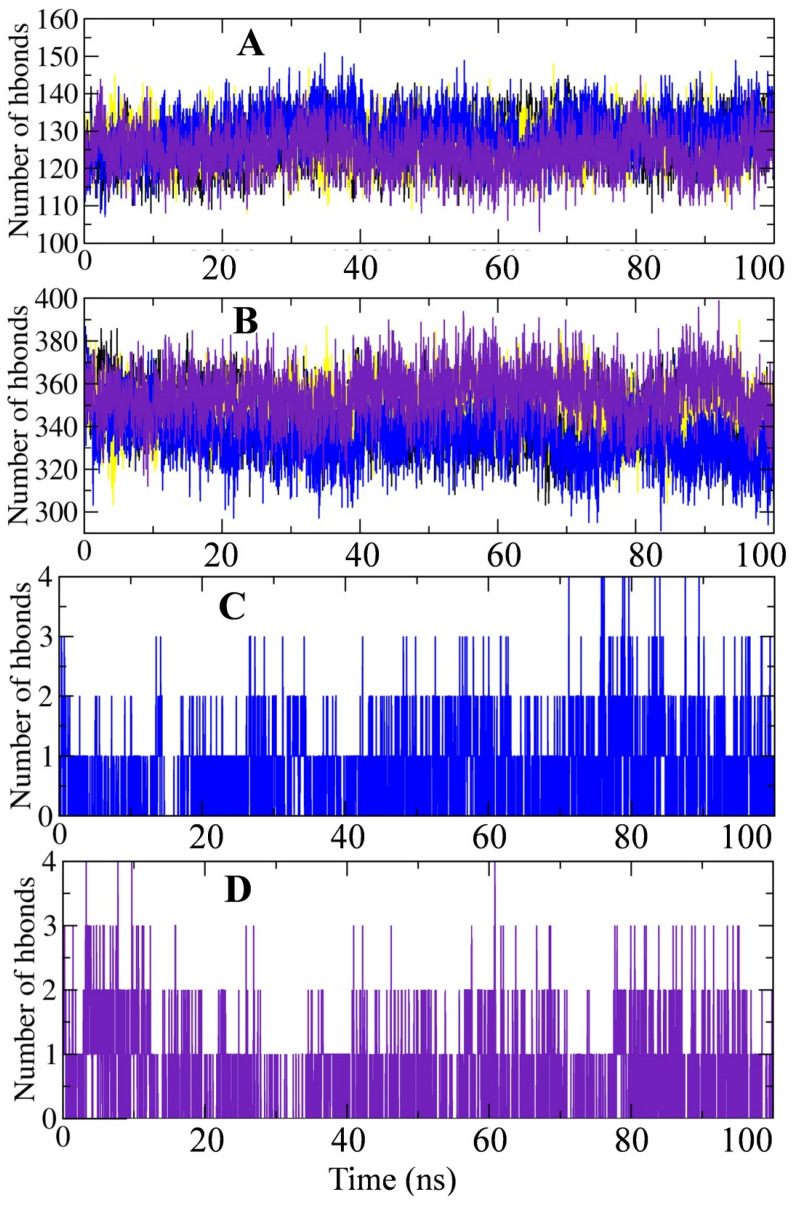
Effect of the M46I mutation on the hydrogen bond formation pattern of saquinavir-bound HIV-1 protease. (**A**) Hydrogen bond formation pattern within the protein, (**B**) hydrogen bond formation pattern between the protein and the surrounding water molecules, (**C**) hydrogen bond formation pattern between the wild-type protease and saquinavir, and (**D**) hydrogen bond formation pattern between the M46I-mutation-carrying protease and saquinavir during the 100 ns simulation. The black, blue, yellow, and purple colors represent the wild-type, saquinavir-bound wild-type, M46I-mutation-carrying protease, and saquinavir-bound mutation-carrying protease systems, respectively.

**Figure 4 viruses-14-00697-f004:**
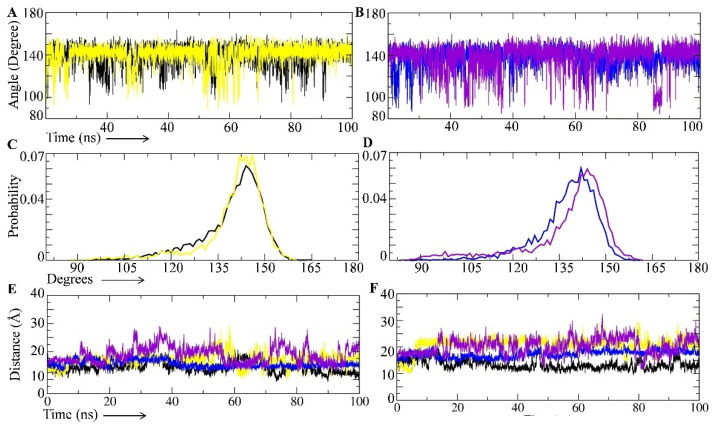
Effect of the M46I mutation on flap curling and the open/semi-open/close conformations of saquinavir-bound HIV-1 protease. The TriCa angle (Gly48–Gly49–Ile50 amino acid residues) distance in (**A**) Saquinavir-unbound wild-type/mutated structure and (**B**) Saquinavir-bound wild-type/mutated structure during the 100 ns simulation period. The TriCa angle degree probability in (**C**) Saquinavir-unbound wild-type/mutated structure and (**D**) Saquinavir-bound wild-type/mutated structure. Fluctuation in the (**E**) Asp25-Ile50 and (**F**) Asp25-Ile149 distances in wild-type, saquinavir-bound wild-type, M46I-mutation-carrying protease, and saquinavir-bound mutation-carrying protease during the simulation period. The black, blue, yellow, and purple colors represent the wild-type, saquinavir-bound wild-type, M46I-mutation-carrying protease, and saquinavir-bound mutation-carrying protease systems, respectively.

**Figure 5 viruses-14-00697-f005:**
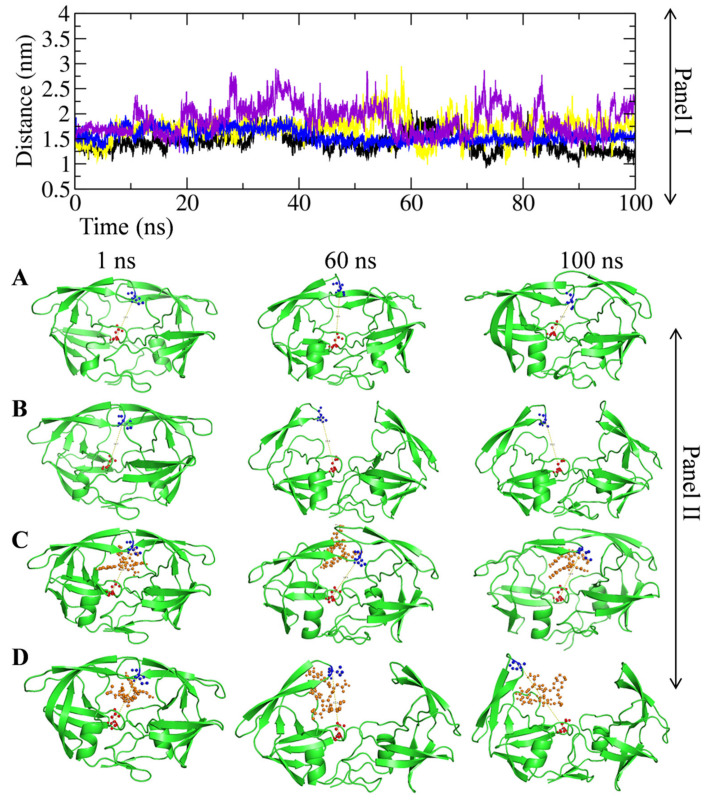
Effect of the M46I mutation on the active site compactness of the saquinavir-bound HIV-1 protease. Panel I shows the change in distance between Asp25 and Asp124 in the WT, SQ-WT, MI, and SQ-MI systems during the 100 ns simulation. The black, blue, yellow, and purple colors represent the WT, SQ-WT, MI, and SQ-MI systems. WT, Wild-type; SQ-WT, Saquinavir-bound wild-type; MI, M46I-mutation-carrying protease; and SQ-MI, Saquinavir-bound mutation-carrying protease. Panel II shows the change in the Asp25 and Asp124 distance values in the (**A**) WT, (**B**) SQ-WT, (**C**) MI, and (**D**) SQ-MI systems at different snapshots (1, 60, and 100 ns) of the MD simulation trajectory. The red and blue colors indicate the Asp25 and Asp124 residues, respectively. The HIV-1 protease and saquinavir structure are shown in green and orange colors, respectively.

**Figure 6 viruses-14-00697-f006:**
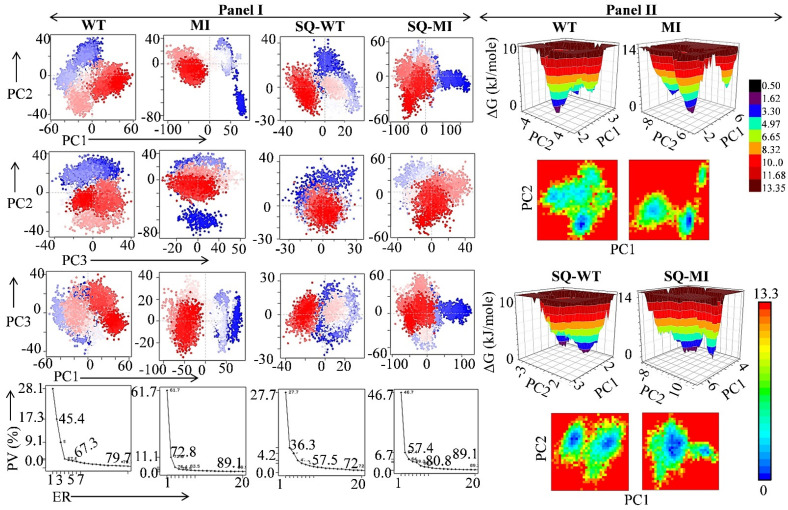
Principal component analysis and free energy landscape of the test systems. (**Panel I**) shows the projection of PC2 vs. PC1, PC2 vs. PC3, PC3 vs. PC1, and the proportion of variance (scree) plot of the WT, SQ-WT, MI, and SQ-MI systems during the simulation period. (**Panel II**) shows the two/three-dimensional free energy landscapes of the WT, SQ-WT, MI, and SQ-MI systems during the simulation period. WT, Wild-type; SQ-WT, Saquinavir-bound wild-type; MI, M46I-mutation-carrying protease; and SQ-MI, Saquinavir-bound mutation-carrying protease; PV, proportion of variance; ER, eigen value rank.

**Figure 7 viruses-14-00697-f007:**
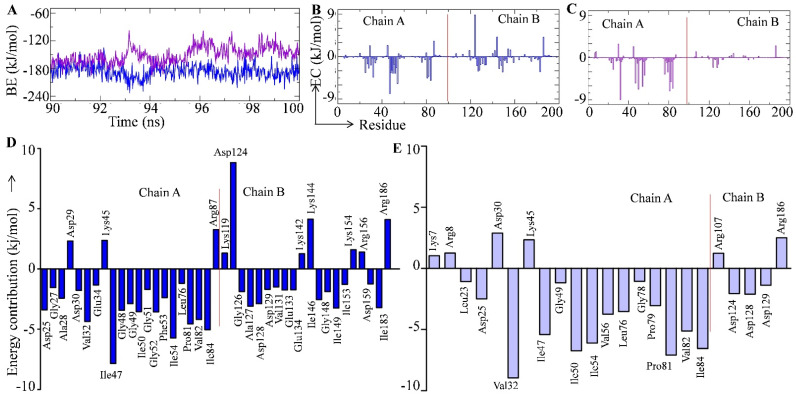
Binding free energy estimation of test complex. (**A**) Binding free energy plot of saquinavir-bound wild-type (blue color) and M46I-mutation-carrying protease (violet color). Decomposition of binding free energy per amino residue in saquinavir-bound (**B**) Wild-type and (**C**) M46I-mutation-carrying saquinavir-bound protease. Residue contribution plot of amino acid residues (<±1 kJ/mol) in (**D**) wild-type and (**E**) M46I-mutation-carrying saquinavir-bound protease.

**Table 1 viruses-14-00697-t001:** Decomposition of binding free energy into its contributors in the test complex.

Type of Energy	SQ-WT Complex (kJ/mol)	SQ-MI Complex (kJ/mol)
van der Waal energy (VdW)	−254.066 ± 13.272	−182.839 ± 17.621
Electrostatic energy (∆G_E_)	−9.563 ± 9.672	−1.634 ± 7.100
Polar solvation energy (PS)	102.179 ± 18.601	51.784 ± 14.940
SASA energy	−24.939 ± 1.587	−17.968 ± 1.858
Binding energy	−186.390 ± 15.282	−150.657 ± 17.875
* Enthalpy (∆H)	−452,079.1 ± 992.6861	−451,974.6 ± 1
* T∆S	−451,892.70	−451,823.94

* Parameters calculated for the entire 100 ns simulation period trajectory.

## Data Availability

Not applicable.

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
