# Peer review of "Drug Resistance Mechanism of M46I-Mutation-Induced Saquinavir Resistance in HIV-1 Protease Using Molecular Dynamics Simulation and Binding Energy Calculation"

_viruses, 2022, doi:10.3390/v14040697_

Round 1
Reviewer 1 Report
This work uses molecular dynamics simulations to study drug-resistance caused by the M46I mutation in the HIV-1 protease. The results indicated that M46I mutation induced structural change. M46I mutation serves as a prototype to provide significant insights for the discovery of inhibitors against drug-resistant HIV-1 protease. The work provided some useful information. While the current manuscript missed formulating specific, experimentally testable predictions. After all, these are the main outcomes of theoretical work. The following comments and questions need to be addressed by the authors.
- Figure B shows the overlapping for wild type and M46I mutation, the change is subtle for the mutation. However, the pose of ligand in mutation is obviously different from it in WT. The protein structure including the active site of the protein was not changed. The score function also suggests a clear difference. This should give a reason.
- The protease system is not a large system. Only 100ns MD simulation is not enough to give a reliable conclusion. A longer MD simulation or multiple replicas are recommended.
- The unit of average binding energy in the SQ-WT and SQ-MI models is kcal/mol in this manuscript, which is not consistent with Table 1 (kJ/mol).
- The units of RMSD,RMSF, Rg, Distance in Figures 2, 4, and 5 should be changed to be in Å. The name of WT, SQ-WT, MI, and SQ-MI systems should be indicated in the upper left or right corner in Figure 2A. The time of Figures S1 and S2 in the supplementary material should be changed to nanosecond precision.
- The calculated binding free energies are much larger than the experimental data, how to explain, how to explain this big discrepancy? Is this attribute to the force field or the method, or both?
- In Figure 7, panels D and E should be combined into one. Instead of marking all residues, the key residues in the test complex, such as Ile47, Asp124 in D and Val32, Ile50, Pro81 in panel E, should be labeled to indicate the contribution of the extra energy to the binding free energy. In Figure 7D, the contribution of Asp124 is positive (~9 kJ/mol), it is obvious that the number of the key residues with the value lowerthan -5 kJ/mol in Figure 7E is more than that of Figure 7D, which seems paradoxical to the results that M46I mutation decreases the interaction between saquinavir and HIV 1 protease. The author should give a reasonable explanation.
- Figure 3is too complex to understand, a large picture is helpful.
- The contribution of ΔHand TΔS should be listed in Table 1, and the symbols are used to represent the type of energy as van der Waal energy, electrostatic energy, Polar solvation energy, etc.
- typos caught:
“kj” in Figure 7D should be changed to “kJ”;
“ij” in formula (2) should be subscript in line 594;
“table” 1 should be “Table” 1 in line 462;
“figure” 4 should be “Figure” 4 in lines 326 and 313.
Line 101, “structur” should be “structure”.
Author Response
- Figure B shows the overlapping for wild type and M46I mutation, the change is subtle for the mutation. However, the pose of ligand in mutation is obviously different from it in WT. The protein structure including the active site of the protein was not changed. The score function also suggests a clear difference. This should give a reason.
Response
We agree with the reviewer that the mutation mediated change in protease structure is subtle, but it induces minor changes in the secondary structure constituting the flap region of the protein which ultimately resulted into the docking score of the compound. The studied mutation is non-active site mutation generally used HIV protease to weaken the drug-protein interaction without affecting the active site (and thus activity) of the protein. The reason has been incorporated in section 2.1.
- The protease system is not a large system. Only 100ns MD simulation is not enough to give a reliable conclusion. A longer MD simulation or multiple replicas are recommended.
Response
We agree with the reviewer comment but due to non-availability of high-end computational facility in our University it is not feasible to run longer simulations or multiple replicas even for such a smaller systems. Moreover, studies on mutation mediated anti-HIV protease drug resistance having ≤ 100ns molecular dynamics simulation periods have been published in much renounced journals recently (DOI: 10.1016/j.jmgm.2017.06.007; doi.org/10.1080/07391102.2018.1492459; doi.org/10.1021/acs.biochem.7b00139; DOI: 10.1016/j.jmgm.2021.107931; DOI: 10.3390/ijms17060819; doi.org/10.1080/07391102.2017.1305296; doi.org/10.1080/1062936X.2021.1999318). This endorses the reliability of the present study.
- The unit of average binding energy in the SQ-WT and SQ-MI models is kcal/mol in this manuscript, which is not consistent with Table 1 (kJ/mol).
Response
The correction have been incorporated in section 2.7, and in the figure 7 A and B.
- The units of RMSD,RMSF, Rg, Distance in Figures 2, 4, and 5 should be changed to be in Å. The name of WT, SQ-WT, MI, and SQ-MI systems should be indicated in the upper left or right corner in Figure 2A. The time of Figures S1 and S2 in the supplementary material should be changed to nanosecond precision.
Response
All the corrections have been incorporated as per suggested.
- The calculated binding free energies are much larger than the experimental data, how to explain, how to explain this big discrepancy? Is this attribute to the force field or the method, or both?
Response
Yes, this attribute is due to the force field and the method both. We have utilized the method of Kumari et al. 2014 doi:10.1021/CI500020M for binding energy calculation. The method development and validation process showed the similar binding free energy of HIV protease with corresponding inhibitor(s).
- In Figure 7, panels D and E should be combined into one. Instead of marking all residues, the key residues in the test complex, such as Ile47, Asp124 in D and Val32, Ile50, Pro81 in panel E, should be labeled to indicate the contribution of the extra energy to the binding free energy. In Figure 7D, the contribution of Asp124 is positive (~9 kJ/mol), it is obvious that the number of the key residues with the value lowerthan -5 kJ/mol in Figure 7E is more than that of Figure 7D, which seems paradoxical to the results that M46I mutation decreases the interaction between saquinavir and HIV 1 protease. The author should give a reasonable explanation.
Response
The Figure 7 B, C, D and E represent the binding of drug with the two chains (A and B) of the HIV protease. The red color line has been incorporated in the figure to distinguish the chains. Normally, drug binds with the both chains of the protein. Figure B and C indicates that due to mutation the drug interaction was increased with the chain A and decreased with the chain B. The difference has been clearly depicted in the Figure D and E which indicates that the very few amino acids of chain B interacting the drug. This might be the reason behind the comparatively more opened structure of the mutated protein and weaker interaction of the drug with the protein during the simulation period. The facts have been incorporated in the section 2.7.
- Figure 3is too complex to understand, a large picture is helpful.
Response
Figure 3 has been restructured and enlarged.
- The contribution of ΔHand TΔS should be listed in Table 1, and the symbols are used to represent the type of energy as van der Waal energy, electrostatic energy, Polar solvation energy, etc.
Response
The contribution of ΔHand TΔS has been listed in Table 1, and the symbols have been used to represent the type of energy as van der Waal energy, electrostatic energy, Polar solvation energy, etc.
- typos caught:
“kj” in Figure 7D should be changed to “kJ”;
“ij” in formula (2) should be subscript in line 594;
“table” 1 should be “Table” 1 in line 462;
“figure” 4 should be “Figure” 4 in lines 326 and 313.
Line 101, “structur” should be “structure”.
Response
Typos have been corrected and checked throughout the manuscript.
Reviewer 2 Report
Rana and his group describe the drug resistance mechanism of M46I mutation induced saquinavir resistance in HIV-1 protease using Molecular Dynamics Simulation and Binding Energy Calculation.
- The study lacks novelty. A recent study of M46I and other mutations with different inhibitors has been published. https://doi.org/10.1186/s12977-020-00520-6. What distinguishes your study from the others?
- Why did the author consider only the M461 mutation with Saquinavir even though the other mutations, N88S, L76V, I50L, and I84V influence resistance and sensitisation towards protease inhibitors? The authors should consider other mutations and protease inhibitors in their study to explore the resistance mechanism.
- Are there open and closed Hiv-protease structures? If so, the authors should also consider them and compare them with their results.
- In the error of GPU computing, the sampling time is insufficient.The sampling time should be increased and replicas should be performed to make the data more conclusive.
Author Response
Rana and his group describe the drug resistance mechanism of M46I mutation induced saquinavir resistance in HIV-1 protease using Molecular Dynamics Simulation and Binding Energy Calculation.
- The study lacks novelty. A recent study of M46I and other mutations with different inhibitors has been published. https://doi.org/10.1186/s12977-020-00520-6. What distinguishes your study from the others?
Response
It has been reported that the background mutations might affect the drug resistance potential of the primay mutation in HIV-1 protease. The study reported by Bastys et al. (2020) studied the effect of M46I and other background mutation(s) on drug resistance potential of N88S and L76V and primary mutations in HIV-1 protease. Based on recently published clinical facts (High abundance or most prevalent mutation associated to saquinavir treatment, and greater transmission potential) on M46I mutation mediated saquinavir resistance in HIV patients we considered only M46I mutation (primary mutation) in the present study.
- Why did the author consider only the M461 mutation with Saquinavir even though the other mutations, N88S, L76V, I50L, and I84V influence resistance and sensitisation towards protease inhibitors? The authors should consider other mutations and protease inhibitors in their study to explore the resistance mechanism.
Response
It has been shown that the occurrence of one mutation in HIV protease might influence the drug resistance potential of another additional mutation in the protein. Sometime it nullifies the effect of any of the two mutations. Recently it has been reported that M46I mutation has been found in high abundance that other known mutations in HIV protease and is one of the most prevalent mutation in HIV protease related to saquinavir treatment and is also possess greater transmission potential. Keeping all these facts in our mind we focused only on saquinavir induced M46I mutation in the present study.
- Are there open and closed Hiv-protease structures? If so, the authors should also consider them and compare them with their results.
Response
Yes, the open and close HIV-1 protease structures are available and we utilized 1TW7 and 4EJK PDBs to calculate the Asp25 β-Carbon and Ile50 α-Carbon distance and compared our result with the same. The information have been incorporated in section 2.4.
- In the error of GPU computing, the sampling time is insufficient. The sampling time should be increased and replicas should be performed to make the data more conclusive.
Response
We agree with the reviewer comment but due to non-availability of high-end computational facility in our University it is not feasible to run longer simulations. Moreover, studies on mutation mediated anti-HIV protease drug resistance having ≤ 100ns molecular dynamics simulation periods have been published in much renounced journals recently (DOI: 10.1016/j.jmgm.2017.06.007; doi.org/10.1080/07391102.2018.1492459; doi.org/10.1021/acs.biochem.7b00139; DOI: 10.1016/j.jmgm.2021.107931; DOI: 10.3390/ijms17060819; doi.org/10.1080/07391102.2017.1305296; doi.org/10.1080/1062936X.2021.1999318). This endorses the reliability of the present study.
Round 2
Reviewer 1 Report
The responses and the revision of the manuscript are satisfactory. It can be accepted now.
Author Response
- The responses and the revision of the manuscript are satisfactory. It can be accepted now.
Response: Thank you for accepting the manuscript.
Reviewer 2 Report
-
- The author should add changes and references to the manuscript so that it will be easy and understandable for readers of the journal. (recently published clinical data) citation
- Again reference is missing, and changes should be made to the manuscript. "Recently, it has been reported that M46I mutation has been found in high abundance that other known mutations in HIV protease and is one of the most prevalent mutation in HIV protease related to saquinavir treatment and is also possess greater transmission potential."
Author Response
- The author should add changes and references to the manuscript so that it will be easy and understandable for readers of the journal. (recently published clinical data) citation
Response: The changes and the associated references have been incorporated as per suggested.
- Again reference is missing, and changes should be made to the manuscript. "Recently, it has been reported that M46I mutation has been found in high abundance that other known mutations in HIV protease and is one of the most prevalent mutation in HIV protease related to saquinavir treatment and is also possess greater transmission potential."
Response: The associated references [8, 9, 11, 12] have been incorporated in the manuscript which were already cited.